

# Robustness of genetic diversity measures under spatial sampling and a new frequency-independent measure

Satoshi Aoki, Fumiko Ishihama and Keita Fukasawa

Biodiversity Division, National Institute for Environmental Studies, Tsukuba, Ibaraki, Japan

## ABSTRACT

The genetic diversity of a taxon has often been estimated by genetic diversity measures. However, they assume random sampling of individuals which is often inapplicable. Except when the distribution of the taxon is limited, researchers conventionally choose several sampling locations from the known distribution and then collect individuals from each location. Spatial sampling is a formalized version of the conventional sampling, which objectively provides geographically even sampling locations to cover genetic variation in a taxon assuming isolation by distance. To evaluate the validity of the spatial sampling in estimating genetic diversity, we conducted coalescent simulation experiments. The sampling locations were selected by spatial sampling and one sample was collected from each location for the sake of theoretical simplicity. We also devised a new measure of genetic diversity, $\varsigma$, which assumes spatial sampling and is independent of allele frequency. This new measure places an emphasis on rare and phylogenetically distant alleles which have relatively small effect on nucleotide diversity. Therefore, it can complementarily serve for conservation studies although it cannot be used to estimate population mutation rate. We compared $\varsigma$ with the other diversity measures in the experiments. Nucleotide diversity, expected heterozygosity and $\varsigma$ showed within 3% relative biases on average while Watterson's theta was 31% overestimation on average. Thus, genetic diversities other than Watterson's theta held good robustness under the spatial sampling.

# INTRODUCTION

Estimating genetic diversity within a taxon is important in conservation and evolutionary biology. Genetic diversity is considered as one of the three components of biodiversity to be conserved in the Convention on Biological Diversity (https://www.cbd.int/). Because estimating genetic diversity is costly, rougher and more inexpensive evaluation indices without using genetic data have recently been suggested (*e.g.*, *Hoban et al., 2020*). However, genetic diversity has been originally represented by measures like heterozygosity (*Nei & Tajima, 1981*) or nucleotide diversity (*Nei & Li, 1979*; *Nei & Tajima, 1981*) which have been developed in population genetics. These measures depend on the allele frequency which assumes random sampling. Random sampling is not always equivalent to haphazard

Corresponding author
Satoshi Aoki, aokis1ll1@gmail.com

sampling which extracts samples without planning or arbitrariness. In random sampling, all samples must have an equal probability to be extracted from statistical populations.

Assuming random sampling in estimating statistics is common in population genetics. Violating this assumption of random sampling can trigger an unexpectable bias in estimation of the statistical parameters, and this means that the current measures of genetic diversity can be imprecise in practice except when the whole population size of the taxon is small enough to enable random sampling. *Tajima (1995)* investigated the effect of non-random sampling on segregating sites, heterozygosity and average number of nucleotide differences. He reported that its effect on the segregating sites was large whereas that on heterozygosity and average number of nucleotides differences was negligible. The sampling scheme in the study (*Tajima, 1995*) was, so to speak, double random sampling, in which sequences were randomly sampled at first, and then the sampled sequences were again randomly sampled. The merits of this sampling scheme are that it can be mathematically analyzed, and the demerits are that it is far from practical procedures.

When one desires to estimate natures of a taxon, the statistical populations are considered to be all individuals of the taxon. If the distribution of the taxon is restricted and all the individuals are easily found, random sampling can be conducted. In the other ordinal cases where the distribution is vast and all the individuals are uncountable, random sampling is difficult to conduct because one cannot assign an equal probability to be sampled to each individual. A two-stage sampling is usually applied to this case. The two-stage sampling of a taxon chooses the multiple sampling locations from the known distribution first and then conducts sampling of individuals in each sampling location (*e.g.*, *Maltagliati et al., 2010*). This is a prevailing sampling method, but is not equivalent to random sampling of individuals.

The conventional two-stage sampling lacks an objective criterion to select the sampling locations. Meanwhile, a kind of sampling methods provides the way to objectively decide the sampling locations. We here call them "comprehensive sampling". Comprehensive sampling aims to select the most diverse set of samples covering the range of the genetic diversity of a taxon. *Quijano, Iriondo & Torres (2012)* created a sampling method which utilized ecological information to cover the diversity. Their method extracts the most diverse sampling locations based on isolation-by-environment, an assumption that individuals at different environments should be genetically different. The spatial sampling by *Aoki & Ito (2020)* extracts the best sampling locations which represent the diversity of a taxon assuming isolation-by-distance (IBD), a theory that geographically closer individuals are genetically closer (*Malécot, 1955*; *Kimura & Weiss, 1964*; *Weiss & Kimura, 1965*). Spatial sampling does not require environmental information, and therefore, it is easily applied to all kind of taxa.

The comprehensive sampling provides objectivity and reproducibility to the conventional sampling method and is expected to collect genetically representative samples of a taxon. However, the bias and accuracy of the genetic diversity estimated through the comprehensive sampling has never been examined by comparing its statistical parameter. The existing genetic diversity measures, such as nucleotide diversity, assumes random sampling, and there is still no measure of genetic diversity assuming the comprehensive
sampling. The purpose of this study is to clarify suitable measure under the assumption of the comprehensive sampling. For this purpose, we quantify the bias and accuracy of the genetic diversity measures estimated under the comprehensive sampling. We also developed a new measure of genetic diversity theoretically applicable to the comprehensive sampling which does not depend on the allele frequencies. We compared this new measure to the existing measures using coalescent simulation data. This study focuses on a measure based on a single random sample per location and spatial sampling. One reason for using a single sample per location is that its simplicity is a good start point to build the theory. Another reason is that sampling one individual per location is the easiest way of random sampling from a location since one does not have to consider correlation of the sampling probability among multiple samples. The reason for using spatial sampling is that it has the basal theory of IBD and easier applicability than sampling using environmental information.

## MATERIALS & METHODS

### Existing and new genetic diversity measures

The expected heterozygosity $H_e$ (*Nei & Tajima, 1981*) is defined as

$$H_e = 1 - \sum_{i=1}^{N} p_i^2, \tag{1}$$

where $N$ is the total number of sequences in a population, and $p_i$ is allele frequency. *Nei (1977)* showed the unbiased estimator of the expected heterozygosity as

$$\widehat{H_e} = \frac{2n}{(2n-1)} \left( 1 - \sum_{i=1}^{n} p_i^2 \right), \tag{2}$$

where $n$ is the sample size.

The nucleotide diversity $\pi$ (*Nei & Li, 1979*; *Nei & Tajima, 1981*) is defined as

$$\pi = \sum_{i=1}^{N} \sum_{j=1}^{N} p_i p_j \pi_{ij}, \tag{3}$$

where $N$ is the total number of sequences in a population, $p_i$ and $p_j$ are allele frequency of the sequence $i$ and $j$, and $\pi_{ij}$ is the substitution rate, or the number of nucleotide differences per nucleotide site between the sequence $i$ and $j$. Clearly, $\pi_{ij} = 0$ when $i = j$. In this measure, the allele frequencies $p_i$ and $p_j$ require the assumption of random sampling for unbiased estimation. To distinguish a statistic and the statistical parameter of nucleotide diversity, we denote the statistic as $\hat{\pi}$, namely

$$\hat{\pi} = \sum_{i=1}^{n} \sum_{j=1}^{n} p_i p_j \pi_{ij}.$$

According to *Nei & Tajima (1981)*, the unbiased estimator $\pi$ for the sample size $n$ is

$$\tilde{\pi} = \frac{n}{(n-1)} \sum_{i=1}^{n} \sum_{j=1}^{n} p_i p_j \pi_{ij}.$$

Watterson's theta (*Watterson, 1975*) is one of estimators of population mutation rate. Therefore, there is no parameter and only an estimator unlike the other measures introduced here. It is defined as

$$\theta_w = K \Big/ \left( \sum_{i=1}^{n-1} 1/i \right),$$

where $K$ is the number of segregating sites.

The new statistical parameter $\varsigma$ that does not rely on random sampling was defined without a frequentistic element like an allele frequency:

$$\varsigma = \begin{cases} \dfrac{1}{M(M-1)} \sum_{i=1}^{M} \sum_{j=1}^{M} \pi_{ij} & (M > 2) \\ 0 & (M = 1), \end{cases} \tag{4}$$

where $M$ is the total number of allele types, or the number of distinct alleles within a taxon. The character $\varsigma$ (final sigma) etymologically comes from "s" of "substitution rate", avoiding $\sigma$ which is often used for standard deviations. The $\varsigma$ ranges from 0 to 1; when a population contains only one allele, $\varsigma = 0$ by its definition, and when all alleles in a population are completely different from each other ($\pi_{ij} = 1$), $\varsigma = 1$. The statistical parameter $\varsigma$ is interpreted as the average substitution rate among all allele types within a taxon. The nucleotide diversity $\pi$ includes the information of the allele frequency, and provides the highest value when all alleles with the same $\pi_{ij}$ have an equal frequency. On the other hand, the average substitution rate $\varsigma$ does not count allele frequencies or the rarity of each allele. Because the new measure $\varsigma$ is not affected by allele frequency which is practically difficult to estimate and rare alleles are expected to be averagely sampled *via* spatial sampling, the new measure has theoretically its advantage in unbiased estimation under spatial sampling. Although the new measure cannot be used to estimate the population mutation rate like the nucleotide diversity, the new measure can serve as a complementary genetic measure of population which emphasizes rare and phylogenetically distant alleles. When all alleles are different from each other ($M = N$, $p_i = 1/N$ for $i = 1$ ... $N$), $\varsigma$ is always larger than $\pi$ and converges to $\pi$ under $N \to \infty$. Therefore, when the observed sequence is long enough to distinguish all the samples and the sample size is large, $\varsigma$ and $\pi$ provide an almost identical value. Concerning $\varsigma$, we can suppose a statistic $\tilde{\varsigma}$,

$$\tilde{\varsigma} = \begin{cases} \dfrac{1}{m(m-1)} \sum_{i=1}^{m} \sum_{j=1}^{m} \pi_{ij} & (m > 2) \\ 0 & (m = 1). \end{cases} \tag{5}$$

Here, $m$ indicates the number of allele types in the sample. Note that when $m = n$, $\tilde{\varsigma} = \tilde{\pi}$. Then, its expected value for $m > 2$ is

$$E[\tilde{\varsigma}] = E\left[ \frac{1}{m(m-1)} \sum_{i=1}^{m} \sum_{j=1}^{m} \pi_{ij} \right]. \tag{6}$$

Formula (6) is the expected value of average substitution rates among all allele types in the population, and this is exactly equivalent to $\varsigma$. Thus,

$$E[\tilde{\varsigma}] = \varsigma$$

However, this estimator assumes that all allele types are randomly sampled, and this assumption is usually not fulfilled by random sampling of individuals of a taxon. Generally speaking, closely related alleles are relatively common in a taxon, and distantly related alleles are rare. Therefore, the estimator $\tilde{\varsigma}$ based on random sampling of individuals is considered to underestimate $\varsigma$ by missing rare distant alleles.

The estimator $\tilde{\varsigma}$ based on random sampling is theoretically inappropriate to estimate $\varsigma$. Then, how about spatial sampling of locations and random sampling of one individual in each location? We use another statistic as the estimator of $\varsigma$. We define the following auxiliary measure

$$\varsigma_n = \max S_n (n = 2, 3, \ldots, M, \ldots, N), \tag{7}$$

where $S_n$ is a set of $\tilde{\varsigma}$s in formula (7) with the sample size $n$. For example, $\varsigma_2$ is $\tilde{\varsigma}$ calculated between the two most distant sequences in the population. Clearly, $\varsigma_N = \varsigma_M = \varsigma$. In some cases, $\varsigma_n$ can be equal to $\varsigma_{n+1}$. For example, when $\varsigma_2$ is calculated from the two sequences 'AAA' and 'GGG' and third sequence is 'TTT', then $\varsigma_2 = \varsigma_3 = 1.0$. Likewise, $\varsigma_4 = 1.0$ when the fourth sequence is 'GGG'. Thus, $\varsigma_n$ decreases or does not change as the sample size $n$ increases. We now prove that $\tilde{\varsigma}$ based on spatial sampling is theoretically equivalent to $\varsigma_n$ in the supporting information (S1). Nucleotide sequences are mathematically too complicated. Therefore, we indirectly treat them as substitution rates in the proof. Discussion about the assumptions will be described in the Discussion section.

Since $\varsigma_n$ monotonically decreases as $n$ increases, $\varsigma_n \geq \varsigma$. When $n \to N$, where $N$ is the total population size, $\varsigma_n \to \varsigma$. Thus, $\tilde{\varsigma}$ based on spatial sampling is also a consistent estimator of $\varsigma$. When all the alleles are different, namely $n = m$, $\varsigma_n = \tilde{\pi}$. Considering from this fact, when $\varsigma_n > \tilde{\pi}$, it means that the frequencies of the alleles which have high substitution rate among them are relatively low, and when $\tilde{\pi} > \varsigma_n$, such allele frequencies are relatively high. Because phylogenetically distant alleles are usually rare, $\tilde{\pi}$ should be larger than $\varsigma_n$ in ordinary cases. The $\varsigma_n$ will be larger than $\tilde{\pi}$ when the species is severely decreasing and its distribution is divided, for example.

The calculation method of $\varsigma$ as a measure of genetic diversity is the same as that of $\pi$. It can be calculated for each locus as well as for the sequences which is made by concatenating multiple loci. Indels in sequences are not directly used to calculate $\pi$ or $\varsigma$, but gap coding (*e.g.*, *Borchsenius, 2009*) enables this as long as its maximum parsimony estimation stands. This method also enables to use microsatellite data to calculate $\pi$ or $\varsigma$.

We now denote the estimated $\varsigma$ as $\hat{\varsigma}$. Simulation experiments are necessary to analyze the effects of the simulation parameters and to compare the estimated $\varsigma$ and $\pi$.

## Estimation test of genetic diversities

First, we define the terms used in the following simulation experiments. Subpopulation means a group of individuals where all individuals randomly breed. A portion of the

individuals migrates to the other subpopulation(s) every generation, and the portion is defined by the migration rate. Population means a group composed of all subpopulations. Meanwhile, statistical population and statistical sample mean data to be estimated and data extracted from the statistical population, respectively. There are two kinds of migration pattern, the stepping stone model and island model. In the stepping stone model, migration occurs only between adjacent subpopulations. The adjacency is judged in a grid manner, and the maximum number of adjacent subpopulations is four. On the other hand, migration in the island model occurs between all the pairs of subpopulations. Variables used in the simulation experiments are described as simulation parameters, whereas the true values of the genetic diversity measures to be estimated are described as statistical parameters.

We start from the explanation of the common part of experiments. The experiments were conducted using coalescent simulations on 10,000 haploid individuals. The coalescent simulation was conducted using ms (*Hudson, 2002*). However, the poor pseudorandom number generator implemented in ms (a linear congruential generator) was replaced by dSFMT which is a descendant of Mersenne twister (*Matsumoto & Nishimura, 1998*). The replaced version of ms is available at https://github.com/heavywatal/msutils. The resultant coalescent trees were processed with Seq-Gen (*Rambaut & Grass, 1997*) to obtain fasta sequence files. One sequence per individual was generated and the sequence length was set to 1,000. The mutation rate per generation followed that used in the comparison of coalescent simulators in *Chen, Marjoram & Wall (2009)*: $2 \times 10^{-8}$/bp. The generalized time reversible (GTR) model (*Tavaré, 1986*) was used for the substitution model. We extracted 10, 20, 30, 40 and 50 sequence samples and calculated $\tilde{\pi}$ for each sample size. The simulation was iterated 20 times for each combination of the simulation parameters, including the sample size. Although $\hat{\varsigma}$ is an unbiased estimator of $\varsigma_n$, it was infeasible to calculate $\varsigma_n$ for each sample size due to combination explosion. Therefore, we compared $\hat{\varsigma}$ to $\varsigma_{10000} = \varsigma$ of which $\hat{\varsigma}$ was a consistent estimator. Likewise, $\pi$ were calculated from each coalescence result using all the 10,000 sequences as the statistical parameter to be estimated by $\tilde{\pi}$. We also calculated expected heterozygosity $H_e$ and per-site Watterson's theta $\theta_w$. The Watterson's theta was divided by 1,000 to obtain the per-site values. The nature of estimators is usually analyzed by multiple data on a single statistical parameter. For example, a bias in the estimation of population mean under random sampling is the difference between the population mean and the mean of data randomly extracted multiple times. However, it is impossible to obtain multiple data on a single statistical parameter in this study, because spatial sampling can provide only a single set of locations and data per simulation. Therefore, instead, we compared the estimators by their bias relative to each statistical parameter. When comparing the mean of the relative biases or the scaled mean error (SME), *e.g.*, the means of $(\hat{\varsigma}_n - \varsigma)/\varsigma$, we checked the sign and size of their relative biases for each simulation result using R (*R Core Team, 2022*). The accuracy or the scaled root mean square error (SRMSE) (*Walther & Moore, 2005*), *e.g.*, the means of $|(\hat{\varsigma}_n - \varsigma)/\varsigma|$ was calculated. The standard deviations of the bias and accuracy were also calculated. An estimator with its relative bias nearer to zero is considered to have smaller bias. If the difference of the accuracy is below zero, it means $\hat{\varsigma}_n$ has better accuracy than $\tilde{\pi}$. Now, we move to the explanation of parts specific to each experiment. In this study,

**Table 1 A list of the simulation experiments.**

| No. | Migration | Subpopulation structure | Subpopulation size |
|-----|-----------|-------------------------|--------------------|
| #1 | Stepping stone model | Designed in a grid manner (A–D in Fig. 1 Left top) | Even |
| #2 | Stepping stone model | Designed in a grid manner (A and C in Fig. 1 Left top) | Uneven (Fig. 1 Right top) |
| #3 | Island model | Random (Fig. 1 Left bottom) | Even |
| #4 | Island model | Random (Fig. 1 Left bottom) | Uneven (Fig. 1 Right bottom) |
| #5 | No migration | Single population | NA |

**Notes.**

NA, Not applicable since there is no subpopulation.

five types of simulation experiments were conducted to evaluate the estimation of genetic diversity measures (Table 1).

The experiments #1 and #2 were based on the stepping stone model and were simulated under even and uneven subpopulation sizes, respectively. The migration rates were 0.001, 0.01, 0.1, 0.5 and 0.9. All adjacent pairs of the subpopulation had the same migration rates. The following four types of stepping stone subpopulation structure were prepared (Fig. 1). The $x \times y$ in the following explanations means the numbers of longitudinal (x) and latitudinal ($y$) subpopulations. A: 100×1 linear structure. B: $100 \times 1$ circular structure. C: $10 \times 10$ plane structure. D: $50 \times 2$ cylindrical structure. These subpopulations were positioned on the artificial coordinates. Since all pairs of the subpopulations had an equal migration rate, coordinates were positioned to have an equal distance between adjacent locations or subpopulations. The distances were calculated in the web site of Geospatial Information Authority of Japan (https://vldb.gsi.go.jp/sokuchi/surveycalc/surveycalc/bl2stf.html), and was based on Geodetic Reference System 1980 (GRS80). $G_{st}$ (*Pons & Petit, 1995*) and $N_{st}$ (*Pons & Petit, 1996*) were calculated for each migration rate to investigate the differentiation among subpopulations. The experiments #1 used all of these subpopulation structures, while the experiments #2 used only the subpopulation structure A and C. Both of the experiments had 100 subpopulations. All the subpopulations in the experiment #1 had an even subpopulation size, namely 100 individuals in each subpopulation. On the other hand, the subpopulation sizes in the experiment #2 followed one-dimensional or two-dimensional truncated normal distribution with the mean zero and the variance one (Fig. 1 Right top). The subpopulation sizes in the experiment #2 were rounded to integers. This distribution of the individuals was intended to follow the center-periphery hypothesis (*Lawton, 1993*), which assumed that individuals at the center of the distribution were abundant and that those at the margins were few. This is the reason for not using the structure B and D that do not have an "edge". Then, using the coordinates of the subpopulations, spatial sampling was conducted using Samploc software (*Aoki & Ito, 2020*) to extract 10, 20, 30, 40 and 50 subpopulations. The simulated annealing in Samploc was repeated 100,000 times, and calculation of the distance was based on GRS80. Finally, one sequence was randomly extracted per location. Mersenne twister was used for the pseudorandom number generator. Namely, we first sampled the

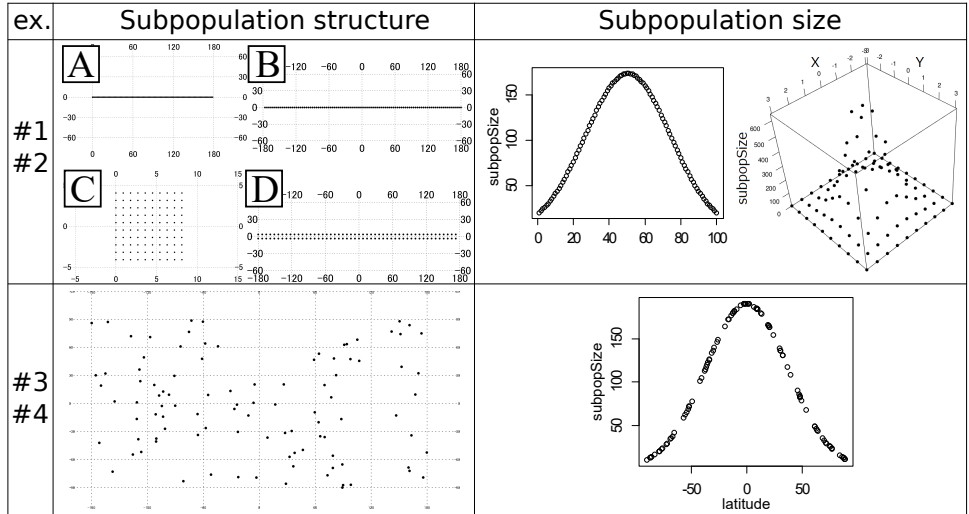

**Figure 1** **Subpopulation structures and subpopulation sizes used in the experiments #1–4.** Note that the subpopulation sizes in the experiments #1 and #3 are even and not shown in the panels. Left top panel: artificial coordinates used for the experiment #1 and #2. (A) 100 ×1 linear structure. (B) 100 ×1 circular structure. (C) 10 ×10 plane structure. (D) 50 ×2 cylindrical structure. Numbers on the figures show the latitude and longitude. Right top panel: Distribution of the subpopulation sizes for the population in the experiment #2. Structure A (linear, left) and C (plane, right). (Left) The vertical axis shows the subpopulation sizes. The horizontal axis shows the relative coordinate of each subpopulation. (Right) The horizontal ($x$ and $y$) axes show the relative coordinate of each subpopulation. The remaining vertical axis shows the subpopulation sizes. Left bottom panel: Randomly generated subpopulation locations for the experiment #3 and #4. Right bottom panel: Relationship between the degree of the latitude (horizontal axis) and the subpopulation size (vertical axis) in the experiment #4. The circles correspond to the subpopulations.

subpopulations in geographically even manners, and then we randomly sampled one sequence per subpopulation.

The experiments #3 and #4 were based on the island model. A total of 100 subpopulations were set in random (uniform) coordinates (Fig. 1 Left bottom). $G_{st}$ and $N_{st}$ were calculated just like the experiments #1 and #2. The latitude and longitude were decided by independent pseudo random numbers based on Mersenne Twister. The migration rate $m_{ij}$ between subpopulation $i$ and $j$ was given by the following formula:

$$m_{ij} = \frac{r}{\overline{d}} \cdot \frac{1/d_{ij}}{1/(\sum_{i=1}^{L}\sum_{j=1}^{L} d_{ij})},$$

where $d_{ij}$ is the distance between subpopulation $i$ and $j$, $\overline{d}$ is the mean of all $d_{ij}$, $r$ is average migration rates, which were 0.001, 0.01, 0.1, 0.5, and 0.9, and $L$ is the total number of subpopulations. The given migration rates decreased as the distance increased in inverse proportion. The distance was calculated based on GRS80. The sample was extracted by spatial sampling in the same way as experiments #1 and #2. The subpopulation size in experiment #3 was even (all 100). On the other hand, the subpopulation size of the

experiment #4 was first approximately given by the next formula:

$$10,000 \cdot dnorm(0.027 \cdot lat_i) \Big/ \sum_{i=1}^{L} dnorm(0.027 \cdot lat_i),$$

where *dnorm(x)* is the probability density function of the normal distribution $N\ (x \mid 0, 1)$, *lat $_i$* is the latitude of the subpopulation and $\sum_{i=1}^{L} dnorm(0.027 \cdot lat_i)$ is the sum of all *dnorm* $(0.027 \cdot lat_i)$. The coefficient 0.027 was decided to make the minimum subpopulation size 10. The obtained relationship between the latitude and the subpopulation size is shown in the right bottom panel in Fig. 1. This formula provides larger sizes to the subpopulations nearer to the equator and smaller to ones nearer to the poles, guaranteeing the total population size 10,000. However, the sizes given by the above formula are not necessarily an integer. Therefore, we secondly rounded down all the sizes to the integer, and then increased each of the subpopulation sizes by one until the sum of the sizes reached 10,000; the increment was preferentially conducted for the subpopulation sizes with the larger rounded-down residuals. The experiments #3 and #4 were intended to avoid the non-uniqueness of the result of spatial sampling on the symmetrical subpopulation structure in the experiments #1 and #2. We will minutely discuss this in the 'Discussion' section.

The experiment #5 was simulated in a single population. There was no subpopulation nor migration. While random sampling is generally inapplicable for wild taxa, it is applicable when the taxon is severely threatened, and its distribution and population size are very restricted. Random sampling is more appropriate rather than spatial sampling, if the distribution is small enough for panmixis where isolation-by-distance cannot be assumed. Testing estimation of $\varsigma$ under random sampling is important to enable comparison of genetic diversities between such a threatened taxon and a non-threatened taxon using the same measure $\varsigma$. Here, we conducted the simulation experiment #5 to test the quality of estimation under random sampling. The samples were extracted from the population by random sampling based on Mersenne Twister.

## RESULTS

First, the statistical population data at the migration rate 0.9 in the experiment #4 contained no mutations. Therefore, its data were excluded from the results. For the same reason, two of the twenty statistical population data were excluded from those in the experiment #5 and in the migration rate 0.5 in the experiment #4.

Since our experiments have numerous resultant data, we first oversee the average result. Figs. 2 and 3 show the mean values of biases and accuracies of the genetic diversities. While the result of the random sampling experiment (#5) showed large difference between $\tilde{\pi}$ and $\hat{\varsigma}$, those of the spatial sampling experiments (#1–4) showed similar values for $\tilde{\pi}$ and $\hat{\varsigma}$. The average values of $\hat{\varsigma}$ were always larger than $\tilde{\pi}$. In most cases, the biases of $\tilde{\pi}$ were slightly higher than those of $\hat{\varsigma}$ except when the low migration rates in the linear subpopulation structure of the experiment #2. The bias of $H_e$ were usually similar to $\tilde{\pi}$ and $\hat{\varsigma}$, and its accuracy was a little better than those of $\tilde{\pi}$ and $\hat{\varsigma}$, but the experiment #5 was

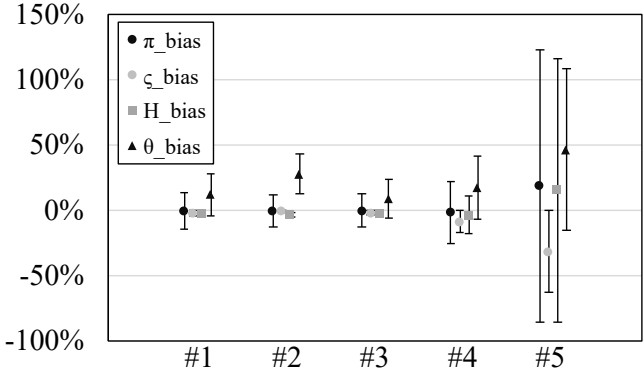

**Figure 2** The relative bias or scaled mean error of estimated genetic diversities in the experiments #1–5.

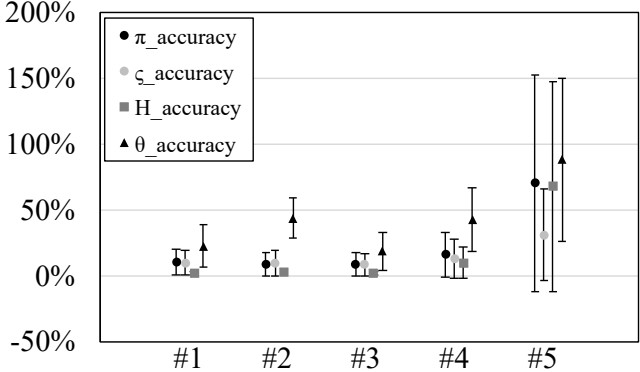

**Figure 3** The accuracy or scaled root mean square error of genetic diversities in the experiments #1–5.

the exception. The estimation of $\theta_w$ was consistently overestimation and showed worse accuracy compared to the other measures. Figure S1 show $G_{st}$ and $N_{st}$ in the tests. The $G_{st}$ mostly increased as the migration rate increased and the $N_{st}$ mostly decreased as the migration rate increased. The exception was the experiments #3 and it showed the converse results. Figure S2 shows individual relative biases and accuracies of genetic diversities. All the biases and accuracies in highly differentiated population (under a low migration rate and a linear, circular or cylindrical subpopulation structure) tended to approach zero as the sample size increased (Fig. S2). On the other hand, the biases and accuracies in population with relatively low differentiation tended to be constant against the increase of the sample size.

## Experiment #1 (designed even subpopulations)

Mostly similar averaged biases and accuracies were observed for $\hat{\varsigma}_n$ and $\tilde{\pi}$, but the standard deviations of $\tilde{\pi}$ were smaller than those of $\hat{\varsigma}_n$ by ca. 1% at the migration rate 0.001 in the subpopulation structure A (linear) and B (circular). The biases of $H_e$ were mostly negative and their accuracies were always better than those of $\hat{\varsigma}_n$ and $\tilde{\pi}$. On the other hand, the

biases of $\theta_w$ were mostly positive and their accuracies were mostly worse than $\hat{\varsigma}_n$ and $\tilde{\pi}$. Especially, the biases of $\theta_w$ at low migration rates were impractically bad. For example, the biases at 0.001 migration rate always exceeded 50%.

### Experiment #2 (designed uneven subpopulations)

All the averaged biases for $\tilde{\pi}$ and $\hat{\varsigma}_n$ mostly showed similar values, but $\tilde{\pi}$ showed 5–10% better biases than $\hat{\varsigma}_n$ at low migration rates in the structure A (linear). Likewise, the standard deviations of $\tilde{\pi}$ were smaller than those of $\hat{\varsigma}_n$ by 3–6% at 0.001 migration rate in the structure A. The biases of $H_e$ were also similar to those of $\hat{\varsigma}_n$ and $\tilde{\pi}$ on average, but the biases of $H_e$ were relatively better than them at low migration rate of the structure A (linear). The biases and accuracies of $\theta_w$ were much worse than the other measures. This tendency was clearer in low migration rates in the structure A (linear).

### Experiment #3 (random even subpopulations)

In most cases, the biases of $\hat{\varsigma}_n$ were similar to those of $\tilde{\pi}$ and were a little lower than those of $\tilde{\pi}$. The biases of $H_e$ were always negative and small. The biases and accuracies were always within 3% at 20 or more sample sizes. The biases of $\theta_w$ was always positive and ca. 10%.

### Experiment #4 (random uneven subpopulations)

Just like the result of the experiment #3, the biases of $\hat{\varsigma}_n$ were similar to those of $\tilde{\pi}$ and were a little lower than those of $\tilde{\pi}$. The biases of $H_e$ were also similar to those of $\hat{\varsigma}_n$ and $\tilde{\pi}$. The accuracies of $H_e$ was about half of those of $\hat{\varsigma}_n$ and $\tilde{\pi}$ except for that of the migration rate 0.5. When the migration rate was 0.5, $H_e$ had the almost same biases as $\tilde{\pi}$, and $\hat{\varsigma}_n$ had the values lower than them. The biases of $\theta_w$ greatly varied depending on the migration rates; they were ca. $-15\%$ for the migration rates 0.001 and 0.01 while they were ca. 15% and 80% for the rate 0.1 and 0.5, respectively.

### Experiment #5 (single population)

The biases of $\hat{\varsigma}_n$ and $\tilde{\pi}$ were not similar and rather, those of $\tilde{\pi}$ and $H_e$ were similar. The biases of $\tilde{\pi}$ and $H_e$ ranged ca. 20–90%. The biases of $\hat{\varsigma}_n$ were ca. $-10$–15%. The biases of $\theta_w$ were ca. 50–90%.

## DISCUSSION

### On the simulation experiments

The $N_{st}$ decreased as the migration rate increased. Since $N_{st}$ reflects differentiation among subpopulations, this behavior of $N_{st}$ is what is expected as more migration usually triggers lower differentiation among subpopulations. On the other hand, the $G_{st}$ showed the opposite tendency. This may be because $G_{st}$ does not consider the substitution rates and does not weight phylogenetically far alleles within a subpopulation which increase at high migration rates or in island model.

The nucleotide diversity, heterozygosity and our new genetic diversity measure $\varsigma$ held within 3% unbiasedness and within 10% accuracy in our simulation experiments using spatial sampling in spite of the violation of the assumption of random sampling. This is

good news for researches and monitoring of genetic diversity, in which almost always the assumption of random sampling is not satisfied. On the other hand, Watterson's theta showed 31% overestimation and 42% accuracy on average. Therefore, its estimation should be avoided in this sampling scheme (spatial sampling of one individual per location).

The smaller values of $\tilde{\pi}$ than $\widehat{\varsigma}_n$ were as expected because this simulation did not shrink the population nor delete the subpopulations. The $\tilde{\pi}$ showed the robustness against the violation of the random sampling assumption under spatial sampling; the bias of $\tilde{\pi}$ was within 2% on average in spatial sampling (the experiments #1–4) and this bias was much lower than the average bias obtained by random sampling (19% in the experiment #5). Considering that the bias and accuracy of $\tilde{\pi}$ behaved similarly to those of $\widehat{\varsigma}_n$, which assumes the comprehensive sampling, we can say that $\tilde{\pi}$ has good robustness against the violation of the random sampling assumption as long as one sample per location is collected under the spatial sampling of the locations. The large positive bias in the experiment #5 may be due to the low statistical parameter; the $\tilde{\pi}$ in the experiment #5 were ca. $10^{-4}$ while those in the experiments #1–4 were ca. $10^{-1}$–$10^{-2}$ except for those at the migration rate 0.5 in the experiment #4 (theirs were ca. $10^{-4}$). The low statistical parameter exaggerates the positive relative biases. In addition, the minimum negative relative bias is $-100\%$ when the statistics is 0 while the positive relative bias can be larger than 100%. This large biases in the experiment #5 were also observed for $H_e$ and $\theta_w$. The bias overestimation of $\tilde{\pi}$, $\widehat{\varsigma}_n$ and $H_e$ in highly differentiated populations (under a low migration rate and a linear, circular or cylindrical subpopulation structure) reflects that spatial samples were more diverse than the statistical population. This tendency should be regarded but should be rather a rare situation. This is because the subpopulation structure of a species is usually plane (structure C in this study) rather than linear, circular or cylindrical (structure A, B or D in this study) and very low migration rates triggers speciation.

The bias of $\widehat{\varsigma}_n$ mostly behaved similarly to that of $\tilde{\pi}$. This is probably because $\widehat{\varsigma}_n = \tilde{\pi}$ when $n = m$. The bias and accuracy of $\widehat{\varsigma}_n$ and $\tilde{\pi}$ in the experiments #1–4 were roughly equal. The accuracy of the estimator $\widehat{\varsigma}_n$ was evaluated by $\varsigma$ not by its statistical parameter $\varsigma_n$ in this study. The standard deviation of the biases were also similar, but the standard deviation of $\widehat{\varsigma}_n$ was slightly larger than that of $\tilde{\pi}$ in some highly differentiated population. This reflects that rare alleles have larger effects on $\widehat{\varsigma}_n$ than $\tilde{\pi}$. Because $\varsigma_n \geq \varsigma$, $\widehat{\varsigma}_n$ should be overestimation as an estimator of $\varsigma$. On average, this was not true for the designed subpopulation in all the experiments. The reason for the underestimation in the experiment #1 and 2 may be imperfection of spatial sampling. Imperfection of spatial sampling is caused by geometric plurality of the best combinations of the locations. For example, the best combinations of the ten locations sampled from the circular subpopulation structure B have ten patterns where every sampled location is ten locations away from the two nearest sampled locations. Even when there are multiple best combinations of the locations for spatial sampling, the best combinations that provides $\varsigma_n$ are not always as many as ones that are best for spatial sampling. Therefore, the geometric plurality decreases the effectivity of spatial samples to estimate $\varsigma_n$. In practical, however, the geometric plurality is rarely problematic. This is because the distance among candidates of sampling locations usually varies, and the candidates are rarely arranged in a geometrically symmetrical manner. The reason of the

underestimation in the experiments #3 and 4 are unknown, but the estimator probably should be adjusted up to obtain the unbiased estimator just as $\tilde{\pi}$ is multiplied by $2n/(2n-1)$ from the statistical parameter even in non-random sampling.

The experiments also showed the robustness of $H_e$ as its biases were within 4%, and this result is consistent with that of *Tajima (1995)*. However, $H_e$ consistently tended to be underestimate. This means that the number of the allele types in the sample are small and that the allele frequencies in the sample are unbalanced compared to those in the statistical population. Because spatial sampling tries to obtain more types of alleles, this underestimated $H_e$ is an unexpected result. Considering the result that $G_{st}$ increased as the migration rate increased, the spatial sampling may do not provide diverse samples when not considering the substitution rate. The accuracy of $H_e$ was smaller than that of $\widehat{\varsigma}_n$ and $\tilde{\pi}$. Unlike the other diversity measures, the biases of $H_e$ in highly differentiated populations were underestimation. The reason for this is unidentified, but maybe the same reason as the general underestimation of $H_e$ discussed above. The reason why the biases of $H_e$ and $\tilde{\pi}$ were almost same in the experiments #5 and at the migration rate 0.5 in the experiment #4 is that almost all the substitution rates between alleles were 1/1,000. In this situation, $H_e$ is ca. 1/1,000 of $\tilde{\pi}$, and so are their statistical parameters. Therefore, their relative biases have almost identical values.

The bias of $\theta_w$ was 34% on average. This tendency of overestimation is as expected because spatial sampling tries to collect more mutations than random sampling. The biases in highly differentiated population were 80–113% even when using 50 samples. Therefore, one probably had better avoid applying $\theta_w$ to species which are expected to be highly differentiated. Otherwise, one should use as many samples as possible to obtain accurate estimates. The biases of $\theta_w$ in highly differentiated population vary more largely depending on the sample sizes than those of the other genetic measures. This is consistent with the known result that the sample size has more effect on $\theta_w$ than $\tilde{\pi}$ (*Subramanian, 2016*).

The samples by spatial sampling can provide more accurate genetic diversity measures than random sampling of a single population. Although the unbiasedness of $\tilde{\pi}$ is theoretically guaranteed when using random sample, the bias in the experiment #5 was larger than those in the experiments #1–4. However, the size of bias does not necessarily deny the unbiasedness. The result should be caused by the fact that the samples by spatial sampling vary little in the twenty times iteration compared to those by random sampling as it is shown in the smaller standard deviation of biases.

## On the new diversity measure, $\varsigma$

The new diversity $\varsigma$ is interpreted as the average substitution rate among all the allele types in a population. As discussed above, its estimator $\widehat{\varsigma}_n$ was estimated as accurate as $\tilde{\pi}$ in most cases. While $\varsigma$ cannot be used to estimate the population mutation rate $\theta$, it can be used to complementarily quantify genetic diversity, regarding rare and phylogenetically distant alleles which do not have much effect on the value of $\pi$. Therefore, $\varsigma$ is appropriate for conservation studies rather than evolutionary studies.

The problem in the theory of $\widehat{\varsigma}_n$ is that the simple IBD assumption in the proof about $\widehat{\varsigma}_n$ is not deductively proved by a population genetic study. This may be due to the difficulty

to treat nucleotide sequences mathematically, and what has ever been proved is only about the relationship between the distance and gene correlation (*Kimura & Weiss, 1964*; *Weiss & Kimura, 1965*). However, the assumption that geographically distant individuals of a taxon are expected to share less nucleotide bases is empirically plausible. Another problem is that we proved the unbiasedness of $\tilde{\varsigma}$ as an estimator of $\varsigma$ under various assumption, but we failed to show its theoretical accuracy, such as mean square errors, relative to $\pi$. Therefore, we could not discuss the resultant accuracy from the theoretical point of view.

Mutations within a location are assumed to have no effect on the substitution rates on average. However, when estimating $\varsigma_n$, one must ideally obtain the best sequence that maximizes $\varsigma_n$ for every location. This means that the ideal method is to collect multiple samples per location, to select the best sample per location and to calculate $\varsigma_n$, where $n$ is not the total number of collected samples but the number of sampling locations. However, conducting this method is difficult due to combination explosion. Even when two samples per location are collected for 50 sampling locations, that requires searching $2^{50} \approx 1.1 \cdot 10^{15}$ combinations.

The genetic diversity $\varsigma$ converges to nucleotide diversity $\pi$ when all allele frequencies are equal, and the number of allele types increases. The effectiveness of $\varsigma$ based on spatial samples varied depending on the dispersal ability of the taxon. When the dispersal ability was low enough to be emulated by the stepping stone model, $\varsigma$ was more unbiased than $\pi$, but $\pi$ was more accurate than $\varsigma$. When the dispersal ability was high enough, $\varsigma$ had better accuracy and almost equal unbiasedness comparing to $\pi$. Therefore, we can say that $\varsigma$ is better than $\pi$ in the island model, but it is not clear that which is better in the stepping stone model. Priority or balance between bias and accuracy may vary depending on the research purpose or the number of the iteration of sampling. Further research is necessary to investigate this problem as well as the statistically strict treatment of bias and accuracy when the statistical parameter is not shared among resampling. Also, this study extracts only one sample per location due to the problem of combination explosion. New theory or calculation method is necessary to utilize information of multiple samples per location for estimation of genetic diversity of a taxon and each location.

## Practical application

According to the result of the simulation experiments, we suggest a practical procedure for applying the tested scheme (Table 2) as follows:

The purpose of this scheme is to estimate genetic diversities of a taxon, which is important for endangered taxa to genetically evaluate the extinction risk (*e.g.*, *Wang & Zhou, 2021*). This scheme is focused on substituting mutations and not on mutations of sequence length like microsatellites. Although microsatellite markers are sometimes employed for genetic assay of endangered taxa, microsatellite markers are known not to accurately reflect genetic diversity in the whole genome (*Väli et al., 2008*). Calculating the new measure $\varsigma$ as well as $\pi$ provides insights into phylogenetically distant alleles. When $\varsigma > \pi$, there are more phylogenetically close alleles than phylogenetically distant alleles. This situation is expected to be ordinary for taxa with a stable or increasing population size without vicariance. However, if $\varsigma > \pi$ with population decrease or vicariance, it means that

**Table 2    Purpose and procedure for applying the tested scheme.**

| Purpose | Estimate genetic diversity of a taxon. |
|---|---|
| Step 1 | Collect distribution data of the taxon. |
| Step 2 | Decide the sampling location by spatial sampling using Samploc[a]. |
| Step 3 | Collect one sample per sampling location. |
| Step 4 | Sequence the samples and obtain an aligned FASTA file. |
| Step 5 | Calculate genetic diversities from the FASTA file using ape[b] and gen.div[c]. |

Notes.
[a] Software (*Aoki & Ito, 2020*). ( https://sites.google.com/view/s-aoki/software/samploc).
[b] An R package (Paradis & Schliep 2019). ( https://CRAN.R-project.org/package=ape).
[c] An R package devised in this study. ( https://github.com/Sa-to-shi-A-o-ki/gen.div_Rpacakge.git).

the taxa is preferentially losing phylogenetically distant alleles or that the subpopulations are young, respectively. The situation of $\pi > \varsigma$ is expected to be ordinary for taxa with a decreasing population size or taxa with vicariance. If endangered taxa are in this situation, conservation breeding should be carefully and rapidly planned to conserve phylogenetically distant alleles before their disappearance by genetic drift.

In the first step, researchers must collect distribution data of the taxon for spatial sampling. Distribution data are geographical coordinates (latitude and longitude) where the taxon was collected or observed. When distribution data is vague (*e.g.*, "collected at Mt. Fuji"), researchers can manually compensate the coordinate of the location. Distribution data can be obtained mainly through GBIF (https://www.gbif.org/) and specimen labels at museums.

In the next step, spatial sampling is conducted to decide the sampling locations using software Samploc (*Aoki & Ito, 2020*). Spatial sampling requires the distribution data, which are collected in the previous step, and the number of sampling locations. The number of sampling locations must be designated by the researcher. More locations will provide more accurate estimation, but more locations cost more time and money. Thus, the number of sampling locations depends on the time and money available for the researchers.

Next, sampling is conducted in the sampling locations decided by the spatial sampling. In this scheme, only one sample per sampling location is necessary, but researcher may keep multiple samples for the other purpose.

At the fourth step, sequencing of the samples is conducted to obtain an aligned FASTA file. We omit the detailed procedure of sequencing and alignment because it depends on the type of sequencing analyses.

At the final step, genetic diversities can be calculated from the FASTA file using R packages ape (*Paradis & Schliep, 1981*) and gen.div. The latter package is what we newly devised because the software we used for the simulation experiments is optimized for simulation experiments and is difficult to apply it to a single set of samples without population data. When multiple samples per location are available, one sample should be randomly resampled for each sampling location. Iterating the resampling provides error bars on the estimated diversities. The FASTA file can be imported into R using read.FASTA

function in ape package. When the alignment includes gaps, mixed bases or unknown bases, deleteGapUnknownColumn function in gen.div function must be applied to remove them. Then, applying calc.diversity function in gen.div package provides seven diversity measures, $H_e$, $\widehat{H_e}$, $\hat{\pi}$, $\tilde{\pi}$, $\theta_w$, per-site $\theta_w$ and $\widehat{\varsigma_n}$. However, please note that Watterson's theta estimated through this scheme has relatively large biases as the result showed.

## CONCLUSIONS

The nucleotide diversity, heterozygosity and our new genetic diversity measure, $\varsigma$, held within 3% unbiasedness and within 10% accuracy in our simulation experiments using spatial sampling in spite of the violation of the assumption of random sampling. This is good news for researches and monitoring of genetic diversity, in which almost always the assumption of random sampling is not satisfied. On the other hand, Watterson's theta showed 31% overestimation and 42% accuracy on average. Therefore, its estimation should be avoided in this sampling scheme (spatial sampling of one individual per location). Our analysis is the first study that tested the performance of genetic diversity measures under comprehensive sampling. The new measure of genetic diversity $\varsigma$ is interpreted as the average substitution rate among all allele types in a taxon, and is not based on allele frequencies. The new measure weighs rare and phylogenetically distant alleles which have less weights in nucleotide diversity but are more important in conservation. By comparing nucleotide diversity and the new measure, researchers can judge whether phylogenetically distant alleles are more abundant than phylogenetically closer alleles.

## ACKNOWLEDGEMENTS

We thank Dr. Yayoi Takeuchi and Prof. Hideki Innan for their advice on selecting the coalescent simulator for this study.

### Funding

This study was supported by JSPS KAKENHI Grant Number JP22J00445. The funders had no role in study design, data collection and analysis, decision to publish, or preparation of the manuscript.

### Grant Disclosures

The following grant information was disclosed by the authors:
JSPS KAKENHI: JP22J00445.

### Competing Interests

The authors declare there are no competing interests.

### Author Contributions

- Satoshi Aoki conceived and designed the experiments, performed the experiments, analyzed the data, prepared figures and/or tables, authored or reviewed drafts of the article, and approved the final draft.

- Fumiko Ishihama analyzed the data, prepared figures and/or tables, authored or reviewed drafts of the article, and approved the final draft.
- Keita Fukasawa analyzed the data, prepared figures and/or tables, authored or reviewed drafts of the article, and approved the final draft.

## Data Availability

The data is available at OSF: Aoki, Satoshi. 2023. "Genetic Diversity without Random Sampling." OSF. August 21. osf.io/n4jpa.

## Supplemental Information

Supplemental information for this article can be found online at http://dx.doi.org/10.7717/peerj.16027#supplemental-information.

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
