# Peer review of "Robustness of genetic diversity measures under spatial sampling and a new frequency-independent measure"

_PeerJ, doi:10.7717/peerj.16027_

## Round 0.1 · original submission · Major Revisions

This paper has received two very thorough reviews. Reviewer 1 is worried that the new statistic will be sensitive to low frequency haplotypes, and encourages the authors to re-frame the paper in terms of the new statistic rather than around random sampling. Reviewer 2 has a number of general concerns, such as the undue focus on pi as the statistic and whether the statistic is appropriate in all contexts. Finally, since you focus on the issue of random sampling, I would like to suggest two references that you do not cite, but probably should be cited (below). I look forward to receiving an extensively revised manuscript.

Tajima, F. (1995). Effect of non-random sampling on the estimation of parameters in population genetics. Genet Res, 66(3), 267-276. doi:10.1017/s0016672300034704

Subramanian, S. (2016). The effects of sample size on population genomic analyses – implications for the tests of neutrality. BMC Genomics, 17(1), 123. doi:10.1186/s12864-016-2441-8

Reviewer 1 ·

Basic reporting

The authors report a novel statistic for haplotype diversity, that is a simpler version of the widely used nucleotide diversity. The authors provide some theoretical results regarding the properties of this new statistic and conduct some simulations of its behavior under evolutionary scenarios.

I am not a native English speaker but I had no issue with the English used in the first sections; however I found the English used in the discussion to be more difficult to understand (e.g., the term “invasion” is used in the discussion where I think the authors mean violation, and there are some errors such as “is currently to be unknown” instead of “is currently unknown”). I am sure the authors could correct it by a careful re-read of the discussion.

Literature references seemed sufficient, although I had some issues with a few statements in the introduction (see below).

Figure quality could be improved. Figure 3 is of low resolution. Can the authors upload a better resolution version? Figures 1 to 5 are simple and methodological figures. Could they rather be regrouped into a single or at most 2 figures with multiple panels?

I think the results themselves and the measure proposed have merit, but I find that the interpretation of the results do not match the results themselves, and I have doubts about some of the methods used which might limit the interpretation of the results (see below).

Experimental design

I think there are several important research question in this paper, mainly the impact of the sampling strategy on estimates of genetic diversity, and the choice of a genetic diversity statistic. Nevertheless, I think that the results and solution proposed by the authors poorly match the main research question they lay out.

The introduction is articulated around the difficulty to perform random sampling, and the methods and results present a new statistic that is supposed to solve these issues. Nevertheless, this new statistic introduces more issues than it solves for this particular problem; indeed, as the authors acknowledge, this new measure is very sensitive to rare alleles and thus more difficult to estimate than nucleotide diversity. Furthermore, it will thus also be very sensitive to population parameters such as mutation rates, and technical parameters such as the size of the genomic region sequenced. Simulation results are not really reassuring in that regard, because they show no real superiority in bias compared to nucleotide diversity, and because they cover a narrow range of parameters that might not be the problematic one (the one leading to many rare alleles). Finally, the absolute value of this new measure is not really shown or discussed, and its interpretation and advantage is unclear. When this new measure and pi are both correctly estimated but disagree (e.g., if one value indicates 0 and the other one 0.5 as in my example below), is it really better for conservation biologists to act using this measure and ignore nucleotide diversity? This should be discussed.

Here are some detailed comments:
3. I do not really agree with the statement that “[genetic diversity] measures depend on the allele frequency which assumes random mating.” Allele frequencies are properties of a population that do not assume anything; for instance, if I were to know the exact proportion p of individuals from a given spatial location that carry allele A, p would be the true allele frequency of A at this location, and H=1-p^2 the exact heterozygosity statistic, whether there is random mating or not. It seems that what is meant is that usual estimators of these quantities assume random mating, but I do not think this is really the case, rather I think it is the theoretical relationship between diversity statistics and population parameters (effective population and mutation for example) that often assumes random mating (e.g., pi = 4Nmu, or the fact that the heterozygosity statistic is really equal to the proportion of heterozygous individuals).

4. What is exactly meant by random sampling and to what extent this is a problem is not clear here. The text seems to imply that in many species, during sampling some individuals are hard to find. Nevertheless, this would only lead to a bias in estimating the frequency of each allele in the case where the probability to be found is associated with the genotype; otherwise, if they are independent, then the individuals found and sampled would still be representative of the whole population. I imagine this is the effect the authors were trying to simulate when using stepping stone models with many subpopulations, which indeed because of isolation by distance could mean that individuals at the periphery, which are probably more genetically distant from the average individual, could be overlooked. Nevertheless, I do not see what is the flaw in nucleotide diversity here: if one samples individuals in all subpopulations, then pi corresponds to the total diversity pi and I do not see how this constitutes a violation of any assumption. Similarly, if some subpopulations have a smaller size and if they are overlooked because of it, it does not mean that the sampling is not random. Actually, the total nucleotide diversity would be correctly estimated as long as the probability of sampling a location and the number of individuals sampled at this location were proportional to its size.


Methods:
5. I am afraid that the measure of diversity proposed, because it does not take into account frequency, will be very sensitive to low-frequency haplotypes. For instance, given 3 haplotypes, where 2 are closely related with pi=0.001, and 1 highly divergent from the rest (pi=0.999) but in reality present in only a single individual of the entire population, the measure would be (0+0.001+0.999+0.001+0+0.999+0.999+0.999+0)/6 = 0.5, which would suggest a diverse population, while pi would close to 0, which is more intuitive. This does not mean that the statistic has no value, but rather that it is complementary to pi rather than a replacement of it, in the same way that the diversity index (which is also independent of the frequency) and the Simpson index (which is not) are complementary. I think this should be acknowledged more explicitely; as it stands it sounds like the new measure could be complete replacement of nucleotide diversity.

6. Another related comment is that as the authors nicely mention, this new statistic, because it is very sensitive to rare haplotypes, will be much harder to estimate from a sample than pi because it will strongly depend on sampling. This results in a paradoxical situation, where the rationale for deriving this statistic (avoiding sampling issues) is at odds with the properties of the proposed solution (a new statistic very sensitive to sampling).

7. Another shortcoming of this new statistic is that it is also probably more sensitive to the size of the genomic region of interest and the mutation rate than pi, because these parameters will increase the number of rare haplotypes which will be hard to capture because of the sampling problem mentioned earlier, so again, I am not convinced of its superiority to nucleotide diversity.

8. l. 200. Do the authors mean the migration rate or scaled migration rate? A migration rate ranging from 0.1 to 0.9 sounds very large for a population of size 10,000, leading to scaled migration rates of 4Nm >= 1000 >>1, which basically means that migration is much stronger than genetic drift so we basically have panmixia and structure (and the migration model) would actually be irrelevant in these simulations. On the other hand, a scaled migration rates ranging from 0.1 to 0.9 is not varying much, and it would be preferable to have values both below 1 (weak migration regime) and above 1 (strong migration regime), for instance 0.1, 1, and 10, corresponding to weak, intermediate, and strong migration. It would be useful to report FST statistics to judge how much the populations sampled are really different.

Discussion:

9. Could the fact that all migration rates used are very high explain that this parameter has no effect on the results?

Validity of the findings

As I mentioned above, I trust that the measure proposed could be useful, and the methods show in a specific scenario that this measure could be estimated correctly. Nevertheless, I do not think the study demonstrates that this measure solves the random sampling issue, nor its superiority to nucleotide diversity.

Additional comments

My suggestion would be to reframe this article around the presentation of this novel statistic, and reducing the emphasis on solving the random sampling problem. I think a good case could be made for this new measure to complement nucleotide diversity, in the same way that species richness and Simpson index can be both used, or that the number of segregating sites and nucleotide diversity are also both used to provide information about a population. Nevertheless, in any case the simulation scenario needs to be improved and more extensive cases should be investigated (other values of M and theta, in particular that leading to high many rare alleles such as weak migration,strong mutation leading to large IBD, in conjunction with small sample sizes), and the usage of this measure should be discussed thoroughly (e.g., I can see how some conservation cases would appreciate a measure that gives a large weight to very divergent, albeit very rare haplotypes, as opposed to nucleotide diversity that gives them a very low weight).

Reviewer 2 ·

Basic reporting

Review of Aoki et al 2022 submitted to PeerJ

Aoki et al present a analysis of one of the standard measures of genetic diversity, nucleotide diversity (π), and compare it to a newly devised measure (c_). I think that it is useful to have an analysis that looks at the effects of sampling scheme on the performance of genetic diversity measures. Overall, the project is a worthwhile contribution to PeerJ, but I think that there are a number of points that should be addressed.

The biggest issue is that I'm not entirely sure that the newly devised measure the authors propose is all that different in concept from Watterson's theta. c_ is basically counting up the number of polymorphisms in a sample and using that as a measure of genetic diversity. That is, in essence, what Watterson's theta does too. Additionally, one of the real benefits of π as a measure is that is directly interpretable in terms of a population's coalescent history, as is Watterson's theta. Is that the case for c_ as well? If it is, it is not clear from the text. Have the authors compared the mathematics of their statistic with Watterson's theta? What about the statistical performance?

Second, on reading this paper, one would likely form the impression that π is the only measure of genetic diversity that is used in the literature. While π may be the most common, it is certainly not the only measure, nor is it always the most appropriate. To adequately compute π, one needs to know how many sites are not variables as well as variable. For certain types of genetic data (such as ddRAD, GBS data or micro-satellites) other measures are more appropriate. I think that the study would be improved by a broader review of the measures of genetic diversity (such as expected heterozygsosity, Watterson's theta, π, Fay and Wu's theta).

Thirdly, the authors should clarify the details of the genetic polymorphims they are talking about. What do the authors actually mean by an "allele"? Modern genetic data almost exclusively deals with single nucleotide polymorphisms (SNPs). So for anyone new to the field, the concept of an allele is entangled with the idea of a SNP. On lines 133-135, the authors are talking about multi-locus alleles - or perhaps insertion/deletion polymorphisms - so it is hard to know exactly what the authors envisage their statistic for. Additionally, while the MS simulator models "alleles" in the loose sense, the simulations model a mutation rate similar to the substitution mutation rate of many eukaryotes, so the vast majority of polymorphisms will have at most 2 alleles.

Fourthly, in many studies that evaluate genetic diversity, a preliminary analysis is to perform an analysis of population structure and to assign samples to discrete populations. Once that is done, diversity is calculated for the different populations on the assumption of random mating (and random sampling) within those clusters. The simulations the authors perform do not include such a step, which is fair enough. However, if they were to sample their simulated sequences from individual sub-populations that might be an approximation of this step. It is not clear from the methods whether the π and c_ that they report are sampled from the meta-population or from individual sub-populations. The authors are right that sampling in empirical studies is not truly random, but I'm not sure if the analysis of their simulations data is truly reflective of how practitioners actually collect and analyze their data.

Minor points

Figures 1, 2 and 3 could be moved to the supplement or merged with Figures 6 and 7 to make the results more impactful.

The part of the discussion talking about environmental DNA is unclear and should either be expanded for clarification, or removed.

Experimental design

N/A

Validity of the findings

N/A

---

## Round 0.2 · Minor Revisions

It looks like reviewer 1 still has some minor issues that they would like addressed. Reviewer 2 is happy with your revision. Once the revised manuscript is provided, I suspect that no more review will be necessary.

Reviewer 1 ·

Basic reporting

The authors did a commendable work to address my comments. I find the additional analyses convincing, and the clarifications regarding the purpose and scope of the findings to be satisfactory. In particular, I appreciate the balanced discussion of the place and usage of the new statistic with regards to alternatives.

Experimental design

No comment

Validity of the findings

No comment

Additional comments

No comment

Reviewer 2 ·

Basic reporting

I have read through the revised manuscript. While I find the text to have been improved, I still have some issues that remain. In particular, I think that the paper needs to address the practical use of the method - how can a practitioner actually use the method, and when should they use it.

In my initial review I wrote:
"... in many studies that evaluate genetic diversity, a preliminary analysis is to perform an analysis of population structure and to assign samples to discrete populations. Once that is done, diversity is calculated for the different populations on the assumption of random mating (and random sampling) within those clusters. The simulations the authors perform do not include such a step, which is fair enough. However, if they were to sample their simulated sequences from individual sub-populations that might be an approximation of this step. It is not clear from the methods whether the π and c_ that they report are sampled from the meta-population or from individual sub-populations."

The authors clarified with:

"As described in the manuscript, the sampling was conducted as follows: “Then, using the coordinates of the subpopulations, spatial sampling was conducted using Samploc software (Aoki & Ito 2020) to extract 10, 20, 30, 40 and 50 subpopulations. The simulated annealing in Samploc was repeated 100,000 times, and calculation of the distance was based on GRS80. Finally, one sequence was randomly extracted per location.” Namely, we first sampled the subpopulations in geographically even manners, and then we randomly sampled one sequence per subpopulation. We added this paraphrase to the manuscript.""

This does make the sampling scheme clearer, but it does not address the first part of my comment. Extracting individuals from across the simulated meta-population, then pooling them without regard to structure in the sampled data is not an accurate reflection of what is done in modern gnomic studies. As I mentioned previously, "a preliminary analysis is to perform an analysis of population structure and to assign samples to discrete populations". Pooling across populations will certainly inflate statistics like Watterson's Theta, as it is sensitive to rare alleles, which will make up a greater proportion of the allele frequency spectrum when sampling stepping stone populations.

A further comment I made was: "... it is hard to know exactly what the authors envisage their statistic for". This was not addressed in the response document. The authors state that their measure would be a complement to conservation studies, but this is not really sketched out in any detail - without a clear set of guidelines for how and when to apply this method as well as a software package or scripts to implement the method, it is hard to know what the authors have in mind.

Minor points.

Figures 2 and 3. It would be much more readable if the authors used the same symbols in their figure legends as are used in the main text.

Increasing point size and readability of Figure would help as well. On the PDF I downloaded, it was quite hard to parse Figure 1 without zooming in.

Experimental design

n/a

Validity of the findings

n/a

Additional comments

n/a

---

## Round 0.3 · accepted · Accept

Thank you for addressing all the comments of the previous two reviews. I now believe that your manuscript is ready for publication. Congratulations!

The Section Editor noted:

> Minor editing, lines 519-521, "This is good news for researches and monitoring of genetic diversity, in which almost always the assumption of random sampling is not satisfied." I am not sure if the intent was "research" or "researchers", but edit accordingly.